# Assessment of Immunogenicity and Neutralisation Efficacy of Viral-Vectored Vaccines Against Chikungunya Virus

**DOI:** 10.3390/v11040322

**Published:** 2019-04-03

**Authors:** César López-Camacho, Young Chan Kim, Joshua Blight, Marcos Lazaro Moreli, Eduardo Montoya-Diaz, Juha T Huiskonen, Beate Mareike Kümmerer, Arturo Reyes-Sandoval

**Affiliations:** 1The Jenner Institute, Nuffield Department of Medicine, University of Oxford. The Henry Wellcome Building for Molecular Physiology, Roosevelt Drive, Oxford, OX3 7BN, UK; cesar.lopez-camacho@ndm.ox.ac.uk (C.L.C.); young@strubi.ox.ac.uk (Y.C.K.); joshua.blight09@imperial.ac.uk (J.B.); marcoslmoreli@gmail.com (M.L.M.); genlalo@yahoo.com.mx (E.M.-D.); 2Division of Structural Biology, University of Oxford, Wellcome Centre for Human Genetics, Roosevelt Drive, Headington, Oxford, OX3 7BN, UK; juha.huiskonen@gmail.com; 3Department of Life Sciences, Imperial College London, Sir Alexander Fleming Building, Exhibition Road, South Kensington, London, SW7 2AZ, UK; 4Virology Laboratory, Federal University of Goiás, Jataí 75801615, Brazil; 5Veterinary Medicine Division, Paul-Ehrlich-Institut, Federal Institute for Vaccines and Biomedicines, Langen, 63225, Germany; 6Institute of Virology, University of Bonn Medical Centre, Bonn 53127, Germany; kuemmerer@virology-bonn.de

**Keywords:** chikungunya virus, CHIKV, ChAdOx1, viral-vectored vaccines, immunogenicity

## Abstract

Chikungunya virus (CHIKV) has caused extensive outbreaks in several countries within the Americas, Asia, Oceanic/Pacific Islands, and Europe. In humans, CHIKV infections cause a debilitating disease with acute febrile illness and long-term polyarthralgia. Acute and chronic symptoms impose a major economic burden to health systems and contribute to poverty in affected countries. An efficacious vaccine would be an important step towards decreasing the disease burden caused by CHIKV infection. Despite no licensed vaccine is yet available for CHIKV, there is strong evidence of effective asymptomatic viral clearance due to neutralising antibodies against the viral structural proteins. We have designed viral-vectored vaccines to express the structural proteins of CHIKV, using the replication-deficient chimpanzee adenoviral platform, ChAdOx1. Expression of the CHIKV antigens results in the formation of chikungunya virus-like particles. Our vaccines induce high frequencies of anti-chikungunya specific T-cell responses as well as high titres of anti-CHIKV E2 antibodies with high capacity for in vitro neutralisation. Our results indicate the potential for further clinical development of the ChAdOx1 vaccine platform in CHIKV vaccinology.

## 1. Introduction

Chikungunya virus (CHIKV) is transmitted by *Aedes* mosquitoes and it is an important growing human health concern in many countries, causing significant outbreaks of acute febrile illness and long-term arthropathy [1]. Its origins are traced to Africa as an enzootic spillover, belonging to the Semliki Forest Complex of the *Alphavirus* (Togaviridae) [2] and was first isolated from a human patient in Tanzania in 1952 [3]. It currently exists as four lineages. The two African enzootic lineages are the West African (WA) and the East Central South Africa (ECSA) lineages. The additional two lineages originated from the ECSA lineage and are called the Asian and the recently emerged Indian Ocean lineage [1,4,5,6]. CHIKV is composed of a positive, single-stranded genomic RNA of 12 kilobases, encoding four non-structural (ns) and five structural (s) proteins [7,8]. The non-structural proteins, nsP1, nsP2, nsP3 and nsP4, are required for virus replication [9]. A sub-genomic RNA encodes the structural proteins: capsid (C), E3, E2, 6k and E1, and thus a polyprotein is produced which is then processed by the capsid auto-proteinase and signalases [10]. The CHIKV surface is mainly composed by E1–E2 heterodimers [11] where E1 glycoproteins mediate fusion [12] and E2 glycoproteins interact with the host receptor [13]. Since its discovery the virus has spread to Asia, Oceania/Pacific, Europe, and mostly recently to the Americas, where there have been more than a million reported cases since its detection in late 2013 [14,15,16,17], whilst major outbreaks continue to be recorded in the Asian and African continents. Although chikungunya vaccines have been under development for several decades, no licensed option is yet available and CHIKV still causes substantial morbidity, overwhelming public health systems and contributing to poverty [18]. In the meantime, current control strategies rely on reducing human exposure to potentially infected mosquito vectors. Several institutions are now engaged in the development of a safe and cost-effective CHIKV vaccine; such vaccines [18,19,20] are based on live-attenuated or inactivated CHIKV, chimeric CHIKV, DNA, subunit, virus-like particle (VLP) and viral-vectored platforms. They are mainly designed to induce humoral responses against the structural viral protein E2, as well as E1, due to strong correlates of protection with neutralising antibodies against structural CHIKV proteins in asymptomatic cases [21]. In addition to that, passive transfer of sera from convalescent humans to mice prevented infection [21] whilst neutralising antibodies against E1 and E2 were able to protect immunocompromised mice [22]. In humans, an early IgG3 neutralising response is associated with reduced clinical symptoms [23] and asymptomatic cases have been associated with high neutralising antibody titres [24]. Here, we describe the design and development of CHIKV vaccines based on the clinically relevant adenoviral vector, ChAdOx1 and the Modified Vaccinia Ankara (MVA) platforms [25,26,27,28,29]. Viral vector expression of the sCHIKV proteins is able to form chikungunya viral particles, thus mimicking a real exposure of CHIKV, whilst being an in silico designed mosaic protein, aiming to represent all four CHIKV lineages. In mice, ChAdOx1 vaccines candidates expressing sCHIKV antigens are able to induce strong humoral and cellular responses upon a single and non-adjuvanted immunisation approach. Importantly, we present evidence of in vitro neutralising activity in sera from vaccinated mice. Finally, whilst durability of humoral responses was achieved upon a single immunisation, MVA vaccines expressing sCHIKV were produced to be used as an alternative heterologous booster regime (ChAdOx1/MVA), in order to increase neutralising antibody titres. Taken together, a viral vectored vaccine for CHIKV, based on the ChAdOx1 platform is a good candidate for further pre-clinical and clinical studies.

## 2. Materials and Methods

### 2.1. Bioinformatics Analysis of CHIKV Genomes and Proteins

Full length structural polyprotein sequences for CHIKV from all lineages were collected from the NCBI protein database (txid37124), aligned using Clustal Omega [30], and a neighbour-joining tree created (Juke-Cantor, 100 bootstraps). Conservation within clades (intra-clade) and subsequently between clades was calculated using our in-house developed software based on a sliding window approach with a sequence weighting method to enable equal representation of all lineages and variants (manuscript in preparation).

### 2.2. Transgene Design

The structural cassette CHIKV sequence derived from various CHIKV lineages was codon optimised. To improve initiation of translation a Kozak consensus sequence was included before the 5∲ end of the transgene. Finally, the transgene design included the required enzymatic restriction sites to allow the in-frame cloning of the transgene between the CMV promoter and the PolyA sequence region contained in our shuttle and expression vector (pMono). A synthetic gene cassette was produced by GeneArt^®^ (Fisher Scientific, Regensburg, Germany) and was named sCHIKV. The sCHIKV plasmid was used as DNA template to further generate the E3,E2,6K,E1 cassette with no capsid by PCR-cloning; this variant was named sCHIKV ∆C. Forward primer: ATGGAGGAATGGTCCCTGGCTATC. Reverse primer: TCATCAGTGCCGGCTGAAG.

### 2.3. Viral-Vectored Vaccine Production

The plasmid containing the sCHIKV cassette (C,E3,E2,6K,E1) and the sCHIKV ∆C (E3,E2,6K,E1) were digested with KpnI and NotI restriction enzymes (NEB, Ipswich, MA, USA) to allow in-frame ligation between the CMV promoter and the Poly(A) regions contained in the shuttle plasmid (pMono). The recombinant DNA plasmids were expanded and purified from *E. coli* using the midiprep kit (Qiagen, Hilden, Germany). Resulting plasmids were verified by restriction analysis and 5∲ and 3∲ flanking sequencing.

To generate ChAdOx1 vaccines, the shuttle plasmids containing attL regions sequences were each recombined with those attR regions contained in the destination vector ChAdOx1 using an in vitro Gateway reaction (LR Clonase II system, Invitrogen^TM^, Carlsbad, CA, USA). Successfully recombined ChAdOx1 sCHIKV and ChAdOx1 sCHIKV ∆C plasmids were verified by DNA sequencing using flanking primers (forward promoter primer and Poly-(A) reverse primer). Standard cell biology and virology techniques were performed to generate the non-replicative adenoviral vectors [28]. To generate MVA based vaccines both the sCHIKV and the sCHIKV cassette ∆C were digested with KpnI and XhoI to allow in-frame ligation between the P7.5 promoter and the TKR locus, contained in the entry plasmid MVA. Ligated DNA plasmids were expanded in *E. coli* and a midiprep (Qiagen) was used for plasmid purifications. Resulting plasmids were verified by restriction analysis and 5∲ and 3∲ flanking sequencing, and co-transfected to produce MVA sCHIKV and MVA sCHIKV ∆C, using the methodology as previously described [31]. Control ChAdOx1 and MVA vectors comprised non-structural (NS) regions from CHIKV and they were named ChAdOx1 NS and MVA NS, respectively.

### 2.4. Animals

Female inbred BALB/c (H-2d), (6–8 weeks) were used for the assessment of immunogenicity (*n* = 6 mice per group). Mice were purchased from Envigo RMS Inc. (Bicester, G.B.). The experimental design took into account the 3R reduction (Replacement, Reduction, Refinement). No randomisation was used in this work.

### 2.5. Immunisation of Mice

For chimpanzee adenoviral-vectored vaccines, mice were vaccinated intramuscularly with a single dose of 1 × 10^8^ infectious units (IU). For boosting, MVA vaccines were administered at a dose of 1 × 10^6^ plaque-forming units (PFU) per mouse. All vaccines were diluted in endotoxin-free PBS.

### 2.6. Cell Culture

Vero cell were grown following the ATCC conditions. HEK-293 cells (ATCC^®^, CRL-1573TM) were grown in DMEM media supplemented with 10% FBS, 1% L-glutamine and 1% non-essential amino acids. Baby hamster kidney-21 (BHK-21) cells were maintained in Glasgow´s Minimum Essential Medium (GMEM) supplemented with 5% fetal bovine serum (FBS), 1% L-glutamine, 10% tryptose phosphate broth, 20 mM HEPES pH 7.2, 100 U/mL penicillin and 0.1 mg/mL streptomycin at 37 °C and 5% CO2. The cell number and viability were calculated by trypan blue staining using the Countess Automated Cell Counter (Life Technologies, Carlsbad, CA, USA). Cell lines are described in the NCBI Biosample database.

### 2.7. Small Scale Transfection and Western Blot Analysis

The expression screening was carried out in Vero cells by transduction of ChAdOx1 sCHIKV, using a MOI = 10. Three days after transduction, the cells and supernatants were harvested, the supernatant was concentrated with a 100 kDa Amicon^®^ spin column (Millipore UK, LTD). The 5x concentrated fraction, the non-concentrated fraction (input) and flow through fraction was also recovered. Cells and supernatants samples were boiled at 100 °C for five minutes in Laemli buffer. Equal amounts of total cell extract and cell supernatants were resolved by SDS/PAGE and transferred to PVDF membranes. Blots were blocked with 1X PBS-Tween- 5% milk and incubated with an anti-CHIKV E1 antibody (AZ 1253, Aalto BioReagents, Dublin, Ireland) at 1:500 dilution, and anti-CHIKV envelope seropositive mice sera (1:500), followed by incubation with HRP-conjugated secondary antibody (1:5000). Chemiluminescence (Perkin-Elmer Life Sciences, Boston, MA, USA) was visualised using the ChemiDoc SRS device (BioRad, Watford H., UK).

### 2.8. Electron Microscopy (TEM)

Formvar/Carbon 200 Mesh Cu grids were glow-discharged in air and loaded with 3.5 µL of the sample from ChAdOx1 sCHIKV vaccine stock or purified VLP of CHIKV by sucrose gradient (10–60%). Excess liquid was removed, and the grids were washed 3 times with MilliQ water. Finally, grids were stained with 2% uranyl acetate for 30 s, excess uranyl acetate was carefully removed using filter paper. The grids were air dried and analysed with a T12 transmission electron microscope (FEI, Eindhoven, The Netherlands).

### 2.9. Ex-Vivo IFNγ ELISpot Assay

ELISpots were carried out using peripheral blood mononuclear cells (PBMCs). Briefly, MAIP ELISpot plates (Millipore UK, LTD) were coated with anti-mouse IFNγ antibody (mAb AN18, Mabtech), after 1h blocking with complete DMEM media (10% FCS). Isolated PBMCs (using ACK buffer solution) were plated alongside with 20-mer specific CHIKV structural peptides overlapped by 10 a.a. (10 μg/mL) and 2.5 × 10^5^ splenocytes from naïve mice per well. After 16 h incubation, cells were discarded, and plates washed with PBS. Following this, 50 μL of biotinylated anti-mouse IFNγ mAb (mAb R4-6A2, Mabtech) (1:1000 in PBS) was added to each well and incubated for 2 h. After washing, plates were incubated with 50 μL of Streptavidin-ALP (Mabtech) at 1:1000 dilution in PBS for 1 h. After another washing step, developing solution (BioRad, Watford H., U.K.) was used. Once spots were visible, the reaction was stopped by rinsing the plates with water. Spots were acquired using an ELISpot reader. Spot Forming Cells (SFC)/10^6^ PBMCs producing IFNγ were calculated.

### 2.10. CHIKV E2 Protein Production 

For expression and purification of the CHIKV E2 protein, the codon-optimised gene of E2 (a.a. 1–346) was cloned into the pHLsec vector [32]. which is flanked by the chicken β-actin/rabbit β-globin hybrid promoter with a signal secretion sequence and a Lys-His6 tag. In order to improve secretion of the E2 protein, the C-terminal region of E2 (a.a. 347–423) was deleted. The pHLsec CHIKV E2 plasmid (500 µg) was transfected in HEK-293T cells using polyethyleneimine (PEI) in roller bottles (surface area of 2,125 cm2) under standard cell culture conditions. Five days after transfection cells were discarded and media was filtered through 0.22 µM disposable filters. The secreted protein was purified from the supernatant by Ni Sepharose affinity chromatography (HisTRAP^TM^, GE Healthcare), using the Äkta Start chromatography system and eluted with Imidazole 500mM. Finally, the eluted protein was dialysed using Slide-A-LyzerTM cassette (Thermo Fisher Scientific, Rockford, IL, USA) against 1× PBS.

### 2.11. Enzyme-Linked Immunosorbent Assay

Antibody binding to CHIKV E2 was measured by an IgG enzyme linked immunosorbent assay (ELISA). Briefly, mice sera were diluted in Nunc Maxisorp Immuno ELISA plates coated with E2 diluted in PBS to a final concentration of 2 µg/mL and incubated at room temperature overnight. Plates were washed 6 times with PBS/0.05% Tween (PBS/T) and blocked with 300 µL with Pierce^TM^ protein-free (PBS) blocking buffer (Thermo Fisher Scientific, Waltham, MA, USA) for 2 h at RT. Mice serum was added and serially diluted 3-fold down in PBS/T with 50 µL per well as final volume and incubated for 2 h at RT. Following washing 6 times with PBS/T, bound antibodies were detected following a 1 h incubation with 50 µL of alkaline phosphatase-conjugated antibodies specific for whole mouse IgG (A3562-5ML, Sigma Aldrich, St. Louis, MO, USA). Following additional 6 washes with PBS/T, development was achieved using 100 µL of 4-nitrophenylphosphate diluted in diethanolamine buffer and the absorbance values at OD405 were measured and analysed using a CLARIOstar instrument (BMG Labtech, Aylesbury, GB). Serum antibody endpoint titres were defined by an absorbance value three standard deviations greater than the average OD405 of the control.

### 2.12. Neutralisation Assay

Titres of neutralising antibodies were determined as described previously using CHIKV replicon particles (VRPs) expressing Gaussia luciferase (Gluc) [33]. Briefly, BHK-21 cells were seeded in 96-well plates at 2 × 104 cells per well. The next day, VRPs (MOI of 2.5) were preincubated with 2-fold serial dilutions of serum samples for 1 h at 37 °C before the mixture was added to the 96-well plates. After incubation for 1 h at 37 °C, the inoculum was removed, cells were washed with PBS, and the medium was added. Readout of secreted Gaussia was performed at 24 h post infection using a Renilla luciferase assay system (Promega, Southhampton, UK). Neutralisation potency was determined as a percentage of measured Gluc activity compared to the Gluc readout after VRP application without serum. Results are presented as 50% neutralisation (NT50) titres.

### 2.13. Ethics Statement

All animals and procedures were used in accordance with the terms of the UK Home Office Animals Act Project License. Procedures were approved by the University of Oxford Animal Care and Ethical Review Committee (PPL 30/2414).

### 2.14. Data Availability

The authors declare that the data supporting the findings of this study are shown in the article.

## 3. Results

### 3.1. Designing of the CHIKV Antigen Cassette

In order to generate a structural gene cassette (C,E3,E2, 6k, E1) from CHIKV, a protein sequence mosaic was constructed using 252 full-length structural polyprotein sequences collected from the NCBI protein database. A neighbour-joining tree was created which identified three distinct clades (A, B, C; Figure 1a). Subsequently conservation was assessed within each clade (intra-clade; Figure 1b) using a sliding window approach, in which a conservation value between 0 and 1 was assigned, 0 being fully conserved. Anything lower than the first quartile (Q1) was classed as conserved. Lastly, the number of windows conserved across each clade and the degree of conservation were identified and used to create a normalised mosaic consensus. Figure 1c illustrates the level of shared conservation between clades, from 0 to -100, with -100 being fully conserved. The resulting sequence mosaic sCHIKV was synthesised (Figure 2a). We constructed chimpanzee adenoviral vector vaccines expressing the full structural cassette (ChAdOx1 sCHIKV), the capsid-deleted structural cassette (ChAdOx1 sCHIKV ∆C) and a non-structural sequence from CHIKV (ChadOx1 NS), the latter being used as a control vaccine (Figure 2b). In addition, Modified Vaccinia Ankara (MVA) viral-vector was used to construct the MVA sCHIKV, the MVA sCHIKV ∆C and the control MVA NS (Figure 2c).

### 3.2. Characterisation of the CHIKV Antigen-Expression

After cloning and production of ChAdOx1 vaccines encoding the genomic sCHIKV cassette, we verified the formation of adenoviral particles by transmission electron microscopy (Figure 3a). Because it has been described that cellular expression of the full structural CHIKV cassette is capable to self-assemble VLPs for CHIKV, as well as for other alphaviruses [34,35,36,37], we collected supernatant from HEK293 cells expressing the sCHIKV antigen. The supernatant was then subjected to sucrose gradient showing a characteristic sedimentation band, which is representative of VLP accumulation (Figure 3b, left). Further electron microscopy preparations revealed particles of approximately 70 nm in size (Figure 3b, right), which resembled wild-type CHIKV particles. Alternatively, we transduced Vero cells with ChAdOx1 sCHIKV and ChAdOx1 sCHIKV ∆C, and verified their expression in cells and supernatants. Specific detection was achieved by western blot, using an anti-CHIKV mice serum and an anti-E1 monoclonal antibody (Figure 3c,d, respectively). In cells transduced with ChAdOx1 sCHIKV, we detected bands at approximately 55 kDa in the supernatants (Figure 3c, lane 1,2,3), with a band of approximately 35 kDa only in the concentrated supernatant (Figure 3c, lane 3). In cellular extracts, we detected specific bands of 35 and 18 kDa (Figure 3c, lane 4). In cells transduced with ChAdOx1 sCHIKV ∆C we also detected bands at approximately 55 kDa in supernatants (Figure 3c, lane 5,6,7), but no bands of 50 kDa or 18 kDa were found in the concentrated supernatant nor in the cell extracts. Therefore, it is suggested that the 35 kDa or 18 kDa band may represent the capsid protein, as previously demonstrated [38], which are not present in ChAdOx1 sCHIKV ∆C-transduced cells. As a positive control, we used a commercially available CHIKV E1 protein (Figure 3c, lane 9); therefore, it is suggested that the 55 kDa band detected in our blots is E1 protein. In addition, a western blot was performed using a monoclonal antibody against CHIKV E1 (Figure 3d); we detected a specific band in the positive control E1 protein (lane 9), as well as a 55 kDa band in all the supernatant fractions from both ChAdOx1 sCHIKV- and ChAdOx1 sCHIKV ∆C-transduced cells. No signal in cell extracts was detected (Figure 3d, lane 4,8), probably due to conformational epitope masking. Taken together, our results suggest that our adenoviral-vectors are able to express the sCHIKV antigens which are able to induce the formation of chikungunya viral-like particles. Thus, we aim to mimic a real antigen exposure of CHIKV particles.

### 3.3. CHIKV-Cellular Responses after Vaccination

To assess the specific immunogenicity of our viral-vectored vaccines, we immunised groups of BALB/c animals (*n* = 6) with a single and non-adjuvanted vaccine dose of ChAdOx1 sCHIKV, ChAdOx1 sCHIKV ∆C, and control ChAdOx1 NS, respectively. Two weeks after immunisation, T cell responses were quantified by ex vivo IFNγ ELISpot, using a specific pool of overlapping peptides spanning the full CHIKV structural cassette, as well as a pool of overlapping peptides related to the control NS vaccines (Figure 4). BALB/c mice receiving vaccines encoding sCHIKV antigens showed high T cell frequencies after stimulation with the sCHIKV peptide pool (ChAdOx1 sCHIKV = mean of 6013 spot-forming cells (SFC)/10^6^ peripheral blood mononuclear cells (PBMCs) and ChAdOx1 sCHIKV ∆C = mean of 6,422 SFC/10^6^ PBMCs); with no statistical difference between both groups, using one-way ANOVA analysis (Figure 4a). The control group ChAdOx1 NS showed low specific T cell responses against the NS peptide pool and no responses to specific sCHIKV peptide pool. In addition, we analysed the responses towards specific peptide sub-pools for C, E3, E2, 6K, E1 and NS (Figure 4b). sCHIKV peptide deconvolution ELISpot allowed the identification of an immunodominant region located in the 6K region which is induced upon ChAdOx1 sCHIKV and ChAdOx1 sCHIKV ∆C vaccination (4,325 and 5,177 SFC/10^6^ PBMCs, respectively). As expected, we detected specific cellular responses towards the capsid in animals vaccinated with ChAdOx1 sCHIKV (mean=984 SFC/10^6^ PBMCs) in comparison with those animals vaccinated with ChAdOx1 sCHIKV ∆C in which low background responses were recorded (Figure 4b). Modest responses were detected in PBMCs against the E2 and E1 peptide pools. Taken together, we conclude that a single injection of ChAdOx1 sCHIKV or ChAdOx1 sCHIKV ∆C is immunogenic and able to induce specific T cell responses in BALB/c mice.

### 3.4. CHIKV-Humoral Responses after Vaccination

To assess humoral responses, 96-well plates were coated with CHIKV-E2 protein. We performed ELISA assays to identify the immunogenic properties of ChAdOx1 sCHIKV and ChAdOx1 sCHIKV ∆C, upon a single and unadjuvanted vaccination strategy (Figure 5A). Two weeks after a ChAdOx1 sCHIKV or ChAdOx1 sCHIKV ∆C vaccination, specific IgG antibody binding to CHIKV E2 protein was detected in sera from vaccinated animals (Figure 5A, left). Six weeks after prime (Figure 5A, right) all the animals increased the levels of OD405 binding to the E2 protein. Of particular interest, 10 months after a single immunisation, sera from vaccinated animals showed specific IgG antibody binding to CHIKV E2, similar to that of six weeks prime group (Figure 5A, bottom). ChAdOx1 NS was used as control as it does not generate antibody responses towards the sCHIKV antigens. Log reciprocal antibody titre analysis (Figure 5b) showed that levels of anti-E2 antibodies increased over time for both ChAdOx1 sCHIKV or ChAdOx1 sCHIKV ∆C prime-vaccinated animals and remarkably they were maintained after 10 months upon a single vaccination strategy. Interestingly, our results indicated that the deletion of capsid in the ChAdOx1 sCHIKV ∆C- vaccinated group did not impair the ability to elicit anti-E2 CHIKV antibody titres. Finally, Modified Vaccinia Ankara (MVA) viral-vector technology has been shown to be a potent booster of immune responses [39,40]. Therefore, we have constructed MVA sCHIKV and MVA sCHIKV ∆C to test their boosting immunogenic potential in a prime-boost heterologous strategy. Mice immunised with either ChAdOx1 sCHIKV or ChAdOx1 sCHIKV ∆C were boosted with MVA sCHIKV and MVA sCHIKV ∆C at six weeks after prime, respectively; and sera was collected two weeks after boost. Prime-boost vaccinated mice showed high endpoint titters of >4 of anti IgG-E2 antibodies (Figure 5b, Prime-Boost section). MVA NS was used as control to boost animals vaccinated with ChAdOx1 NS vaccines. Taken together, we conclude that a single vaccination with no adjuvant strategy is able to induce early, specific and long-lasting B responses in BALB/c mice towards CHIKV-E2 protein.

### 3.5. CHIKV-Neutralising Capacity in Vaccinated-Mice Sera

We tested BALB/c mice sera obtained after an early prime time-point and in a prime-boost vaccination to assess CHIKV antibody neutralisation activity in vitro. The method we used is based on the virus replicon particle-based (VRP) chikungunya virus neutralisation assay to determine NT50 values [33]. Naïve and control ChAdOx1 NS sera were included alongside the assay. At two weeks post-vaccination, sera from the ChAdOx1 sCHIKV- and ChAdOx1 sCHIKV ∆C-vaccinated groups showed high neutralisation activity against VRP CHIKV with a NT50 titre of 5.39x10^3^ and 3.18 x10^3^ respectively (Figure 6a). In a prime-boost regime (Figure 6b), ChAdOx1 sCHIKV/MVA sCHIKV showed a further 2.8-fold increase of the NT50 titter (1.53x10^4^), in comparison to the prime vaccination regime of ChAdOx1 sCHIKV. A similar increase of NT50 titre was found in the sera from mice vaccinated with a prime-boost ChAdOx1 sCHIKV ∆C/MVA sCHIKV ∆C, with a 6-fold increase (1.87x10^4^), when compared to that of the NT50 titre from single ChAdOx1 sCHIKV ∆C vaccinated animals (Figure 6b). These results indicate that when the capsid is deleted from the CHIKV structural genes, a decrease in the antibody NT50 titre is observed, upon a single ChAdOx1 vaccination. Conversely, when a prime-boost regime is used, the deletion of the capsid is beneficial to increase the CHIKV NT50 antibody titres. Taken together, our results confirm that our viral-vectored vaccines elicit functional neutralising antibodies against CHIKV-infective particles, tested by an in vitro model.

## 4. Discussion

In this work we have constructed CHIKV vaccines based on viral-vector platforms, such as ChAdOx1 and MVA vectors. The immunogen was designed based on a mosaic consensus sequence from the Asian, East, Central and South African and West African regions. Although individual strains are antigenically related and CHIKV vaccines previously reported have been developed using heterologous strains [34], a mosaic-based approach ensures a high coverage aimed at raising neutralising antibodies against all variants circulating worldwide. Here, we have developed two different vaccine immunogen candidates: a vaccine encoding the full structural CHIKV polyprotein (sCHIKV) and a vaccine in which the capsid antigen was deleted (sCHIKV ∆C). One of the functions of the capsid is to auto-cleave from the E3-E2-6K-E1 polyprotein. This proteolytic process exposes a signal leader which is contained in the amino-terminal region of E3 and promotes the entry of the polyprotein into the endoplasmic reticulum [41]. There are several advantages in comparing these two different antigenic cassettes. Genetic vaccines, such as DNA vaccines, RNA vaccines, and viral-vectored vaccines, are often limited by the number of nucleotides that can be inserted into the antigenic cassette. Exploring the minimum sequence length able to induce immunogenicity and efficacy (either by in vitro neutralisation or by sterile protection after a viral challenge), will inform the vaccine development of new generation of vaccine candidates against CHIKV. In addition, optimising the minimum number of genetic antigens could allow the inclusion of further antigens to create combined vaccines (either homologous or heterologous) to induce immune response against different pathogens, all in a single genetic vaccine approach. Based on this assumption, efforts are underway to create ChAdOx1 vaccines expressing shorter versions of the CHIKV polyprotein, such as E2 and E1 proteins from CHIKV and other alphaviruses. E2 protein is involved in receptor recognition, while E1 protein is involved in membrane fusion [12,13]. Therefore, exploring single-protein CHIKV vaccines in ChAdOx1 or in other viral-vectored platforms seems feasible and promising, since anti-E2 and anti-E1 antibodies are high in convalescent CHIKV-immune populations [21]. However, ensuring proper antigenic folding in such developments, identical or similar to those antigenic structures in virions from CHIKV is paramount. Here, we have demonstrated by transmission electron microscopy, that our adenoviral vaccine ChAdOx1 sCHIKV is capable of expressing chikungunya-VLPs, which promotes the correct conformational antigens to be presented upon vaccination, therefore mimicking a real exposure to wild-type CHIKV. It will be interesting to explore and compare the neutralisation efficacy between encoded chikungunya-VLPs, E2 or E1 proteins in viral-vectored vaccines.

In this work we have tested the immunogenicity of our viral-vectored vaccines by assessing the cellular and humoral responses in BALB/c mice. Our data shows that the T cell immunogenicity elicited by ChAdOx1 sCHIKV and ChAdOx1 sCHIKV ∆C is directed primarily to a peptide region located in the 6K protein, by IFN-γ production. It will be interesting to explore the development of T-cell based vaccines against CHIKV. The role of T cell immunity in alphavirus infection, especially in CHIKV is not very well understood. Importantly, antigen design and the modulation of its intracellular expression will play a major role at inducing T cell responses directed to the antigen. A clear example is the comparison between CHIKV and CHIKV ∆C ELISpots, in which T cell responses directed to capsid were detected in the CHIKV groups but it was ablated in the CHIKV ∆C groups. It will be interesting to explore the protective and immunological relevance of the T cell response directed by the capsid in pre-clinical models. Although we identified 6k as the immunodominant region in BALB/c, this observation may differ in other inbred or outbred mouse strains; or in the case that vaccines comprise only E3, E2 or E1 antigens. For example, CHIKV-specific CD8+T cells were directed mainly against E1 and E2 proteins in C57Bl6 mice [42,43,44]. T cells induced after CHIKV infection in C57BL/6 mice were not essential to control viremia but CD4 T cells were involved in CHIKV pathology [45]. Furthermore, T cell adoptive transfer did not confer protection against CHIKV challenge [46]. Finally, T cell responses in human CHIKV infection is also poorly understood. T cell responses (CD8^+^ > CD4^+^) have been found in chronic and convalescent CHIKV patients [47], in which specific responses were detected for E2, capsid and ns protein 1 (nsP1). Further studies using CHIKV challenge models would be informative to further dissect the role of cellular responses in protective efficacy using ours or other vaccine platforms.

In terms of humoral responses, we have assessed the binding of antibodies from vaccinated mice by ELISA, coating with E2 protein. E2 protein has been shown to be a primary target for neutralising antibodies, since E2 functions for recognition and attachment to cellular receptors. Here, we have demonstrated that our vaccines are able to mount anti-E2 antibody responses. The level of antibody responses after a single immunisation of ChAdOx1 sCHIKV or ChAdOx1 sCHIKV ∆C were maintained for up to 10 months, highlighting the suitability of our vaccine approach to promote long-lasting immune responses against CHIKV. Nonetheless, we have tested a boosting strategy using MVA chikungunya vaccines to further increase the anti-E2 antibody titres.

Although we have used an ELISA assay to detect anti-E2 antibodies, we are possibly detecting only antibodies against monomeric E2. However, our vaccine platform assembles VLP for CHIKV, in which conformational epitopes might be physiologically relevant to confer protective efficacy in pre-clinical models. Finally, we do not rule out the possibility of inducing responses other structural components.

The ChAdOx1 sCHIKV and the ChAdOx1 sCHIKV ∆C prompt responses in both, B- and T-cell compartments. In vitro neutralisation assays are an excellent standard to test the capacity of CHIKV sero-positive serum to neutralise CHIKV. Here we utilised a widely used neutralisation assay, using a non-infectious CHIKV particle which is fully capable of packing CHIKV replicon RNA, which carries the Gaussia Luciferase (Gluc) gene located in the subgenomic RNA region [33]. Therefore, infected cells secreting Gluc can be detected as a surrogate detection of CHIKV infection. In this study, we tested a rather early time-point after a ChAdOx1 sCHIKV or a ChAdOx1 sCHIKV ∆C prime vaccination, with the hope to demonstrate efficacy of neutralisation after two weeks post-vaccination, as vaccines should be aimed to guarantee protection in a shortest window of time between vaccination and pathogen exposure.

Conversely, we have tested the neutralisation capacity upon a booster vaccination using MVA sCHIKV and MVA sCHIKV ∆C, respectively. Our results demonstrate that our ChAdOx1 CHIKV and ChAdOx1 CHIKV ∆C were able to induce >10^3^ NT50 titres. Since the NT assay presented here has been used in several vaccine development studies for CHIKV, we are able to afford a comparative assessment between our NT50 titters and other vaccine developments. Garcia-Arriaza et al. reported a MVA-CHIKV vaccine expressing the full structural cassette, in which the mean of NT50 titters were <10^3^ in a 6-week prime post-immunisation strategy [44]. Hallengärd et al., developed an attenuated CHIKV vaccines in which they achieved >10^4^ when using the vaccine ∆5NsP3 [48,49]. Finally, CHIKV neutralisation titres have been assessed in human sera yielding NT50 titres of 2.6x10^4^, 1.8x10^4^ and 1x10^4^ in three different CHIKV patients returning from Mauritius, La Reunion and Seychelles, [33,50] respectively. Therefore, the mean of the NT50 titres we have achieved in both prime (>10^3^) and prime-boost regimes (>10^4^) reflect the capacity of our viral-vectored vaccines to induce physiologically relevant NT50 titres.

No licensed vaccine is yet available to prevent CHIKV infection. From >20 experimental vaccines, only 4 have reached clinical trials and three are still active and leading the field [18]. A leading vaccine consists of a virus-like particle that presents antigens similar to CHIKV but without replicating, stimulating immune responses but requiring multiple injections to induce antibody titres [51], thus increasing costs and logistics for its use in in low-income countries. Another development consists of a recombinant live attenuated measles virus (MV) expressing surface CHIKV antigens [52]. The vaccine was immunogenic and safe in humans, but it also requires >2 doses. Regarding adenoviral vectors, only one human adenovirus expressing CHIKV structural antigens has been published [35]. Unfortunately, human adenoviral-vectored vaccines are not suitable to vaccinate humans, due to the widespread presence of wild-type adenovirus eliciting anti-adenovirus immunity and thus decreasing vaccine efficiency [53]; hence the use of chimpanzee adenovirus circumvent this problem.

In the ideal CHIKV vaccine development, safety and immunogenicity should be achieved by a single vaccine dose and no adjuvant requirement. Such a simple and robust development could benefit low-income settings, in which affordability and simplification of logistics are essential to deliver the vaccine to a target population. However, in the event that a booster vaccine is needed, an MVA-based vaccine for CHIKV could be used. Given the suitably of ChAdOx1 in the clinical setting, the capability to produce millions of doses in a cost-effective background, its immunogenic and safety profile in humans, and its suitability to be produced under good manufacturing practices (GMP); ChAdOx1 sCHIKV vaccines are relevant candidates to be further explored in pre-clinical and clinical trials.

## Figures and Tables

**Figure 1 viruses-11-00322-f001:**
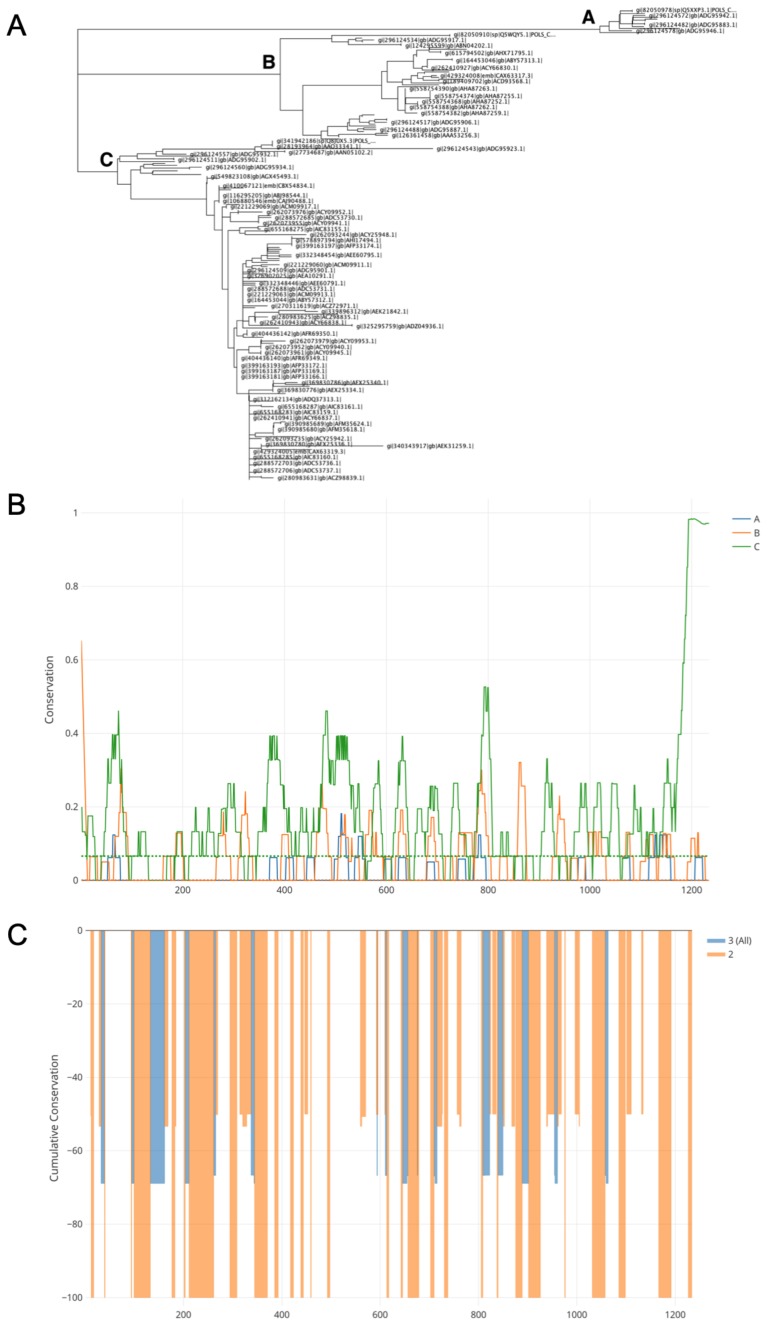
Design of mosaic chikungunya virus (CHIKV) structural polyprotein sequence. (**A**) Neighbour-joining tree of full-length CHIKV structural proteins. (**B**) Intra-clade conservation (**A**–**C**) assessed by in-house sliding window algorithm. Conservation scaled between 0 and 1 with 0 being fully conserved within respective clades. Q1 cut-off for conservation of each clade indicated by coloured dashed line. (**C**) Inter-clade conservation (between A, B and C) across the three clades. Degree of conservation between either all three clades (blue) and only two clades (orange) indicated by cumulative conservation between 0 to −100.

**Figure 2 viruses-11-00322-f002:**
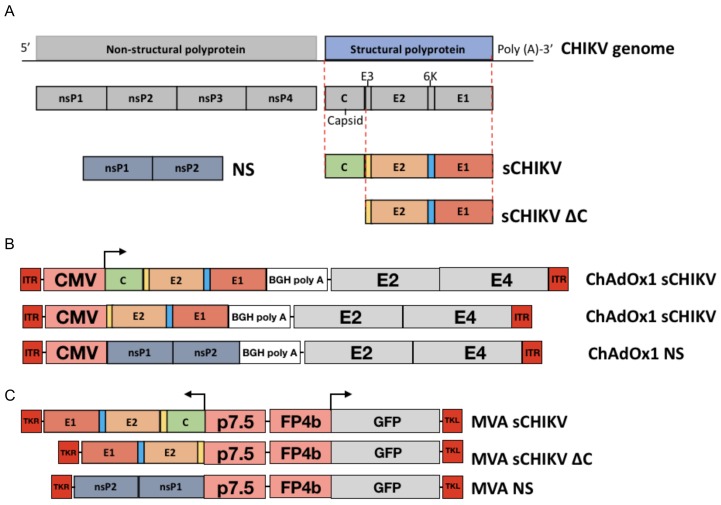
Genome organisation of CHIKV and generation of vaccine candidates. (**A**) Schematic figure of the full genomic RNA of CHIKV depicting the non-structural and structural coding regions. Full structural antigens (sCHIKV), capsid-deleted structural antigens (sCHIKV ∆C) and non-structural antigens (NS) from CHIKV were synthesised. (**B**) ChAdOx1 vaccines were produced to encode the stated candidate genes under the regulation of the cytomegalovirus promoter (CMV). The antigen cassette is located within the Early gene E1-deleted region. Grey boxes are a graphical representation of the adenoviral genome, with the E3 deletion. ITR (inverted terminal repeat), Bovine growth hormone (BGH) poly adenylation region. (**C**) Modified Vaccinia Ankara (MVA) vaccines were produced to encode the stated candidate genes, driven by the p7.5 promoter. Marker GFP is driven by the FP4b promoter. Antigenic cassette is inserted between the thymidine kinase (TK) locus. Non-structural regions were used to generate the control vaccines ChAdOx1 NS and MVA NS.

**Figure 3 viruses-11-00322-f003:**
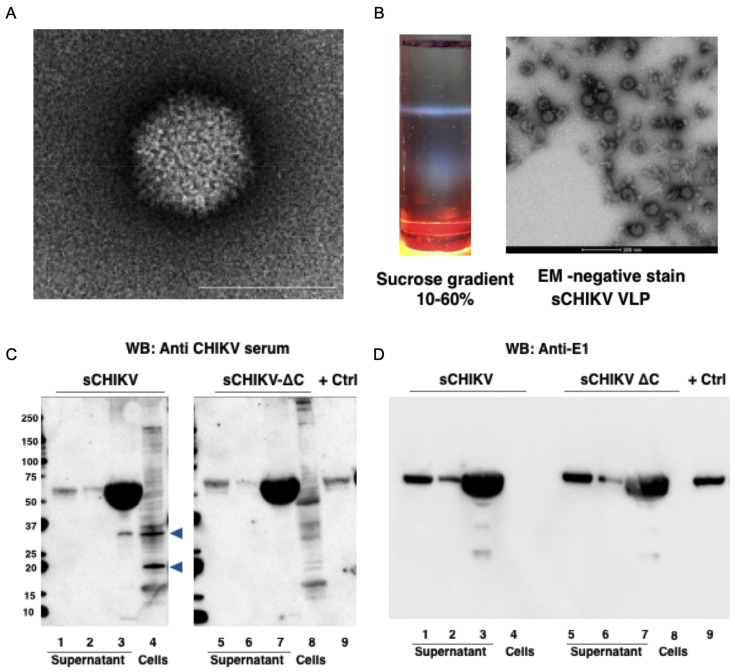
Characterisation of recombinant chimpanzee adenoviral (ChAdOx1)-based vaccines for CHIKV. (**A**) Electron microscopy from a ChAdOx1 sCHIKV vaccine preparation, white line is 100 nm. (**B**) Sucrose gradient purification from the supernatants after expression of CHIKV structural cassette (left). Transmission electron microscopy showing CHIKVLP formation from the isolated gradient band, the white line is 200 nm. (**C,D**) Transduction of ChAdOx1 sCHIKV and ChAdOx1 sCHIKV ∆C in Vero cells; supernatant and cellular extracts were probed against anti-CHIKV serum (**C**) or anti-E1 monoclonal antibody (**D**). Blue arrowhead in sCHIKV blot shows bands that were not present in sCHIKV ∆C.

**Figure 4 viruses-11-00322-f004:**
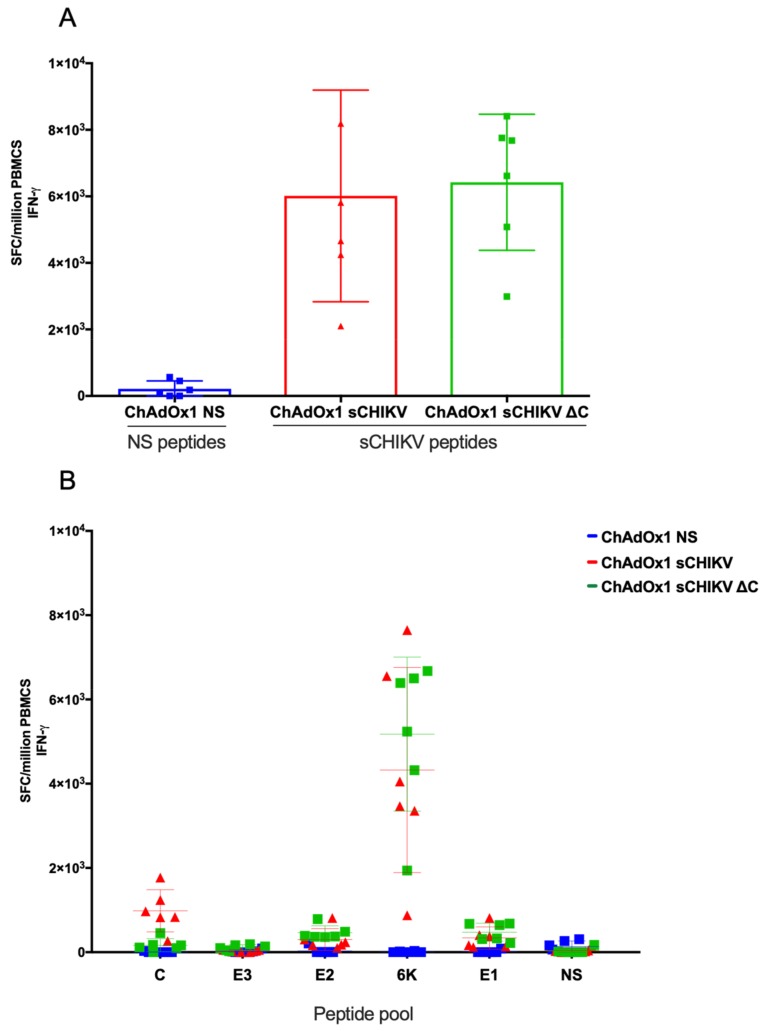
Cellular immune responses elicited by ChAdOx1 vaccines. (**A**) BALB/c mice (*n* = 6 per group) were immunised intramuscularly (i.m.) with a dose of 10^8^ I.U. of ChAdOx1 vaccines. Peripheral blood mononuclear cells (PBMCs) were cultured with a peptide pool containing the CHIKV structural antigens by ELISpot for IFNγ producing cells. (**B**) ELISpot from the same mice, using deconvolution of peptides spanning the C, E3, E2, 6k and E1 regions, respectively. Values represent the spot-forming cells (SFC) per million PBMCs. 20-mer peptides spanning the CHIKV structural polyprotein (10 µg/mL) were used for stimulation. Line colours and shapes represent mice vaccinated with each vaccine.

**Figure 5 viruses-11-00322-f005:**
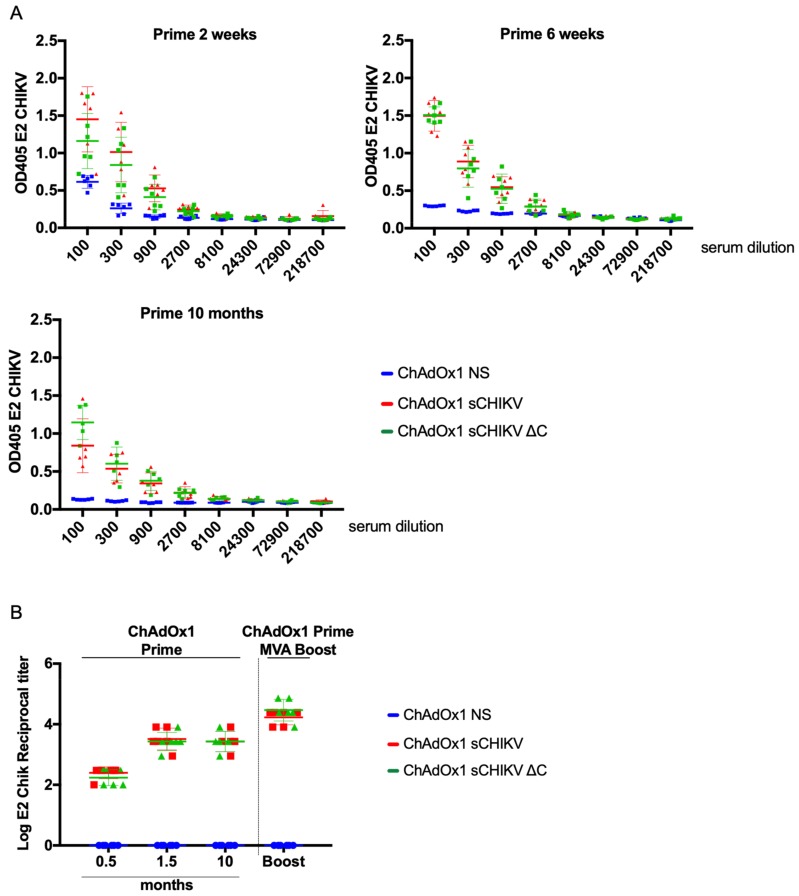
Humoral immune responses elicited by viral-vectored vaccines. (**A**) BALB/c mice (*n* = 6 per group) were immunised intramuscularly (i.m.) with a dose of 10^8^ I.U. of ChAdOx1 vaccines. CHIKV E2 ELISA was performed with sera obtained from vaccinated mice at week 2, week 6 and 10 months after a single ChAdOx1 vaccination in which the OD405 values are represented over several 3-fold sera dilutions. (**B**) Reciprocal log E2 ELISA titters were calculated for all groups shown in A. In addition, a heterologous prime 1.5 month-boost 0.5 month is presented. Colours and shapes represent groups vaccinated with each vaccine.

**Figure 6 viruses-11-00322-f006:**
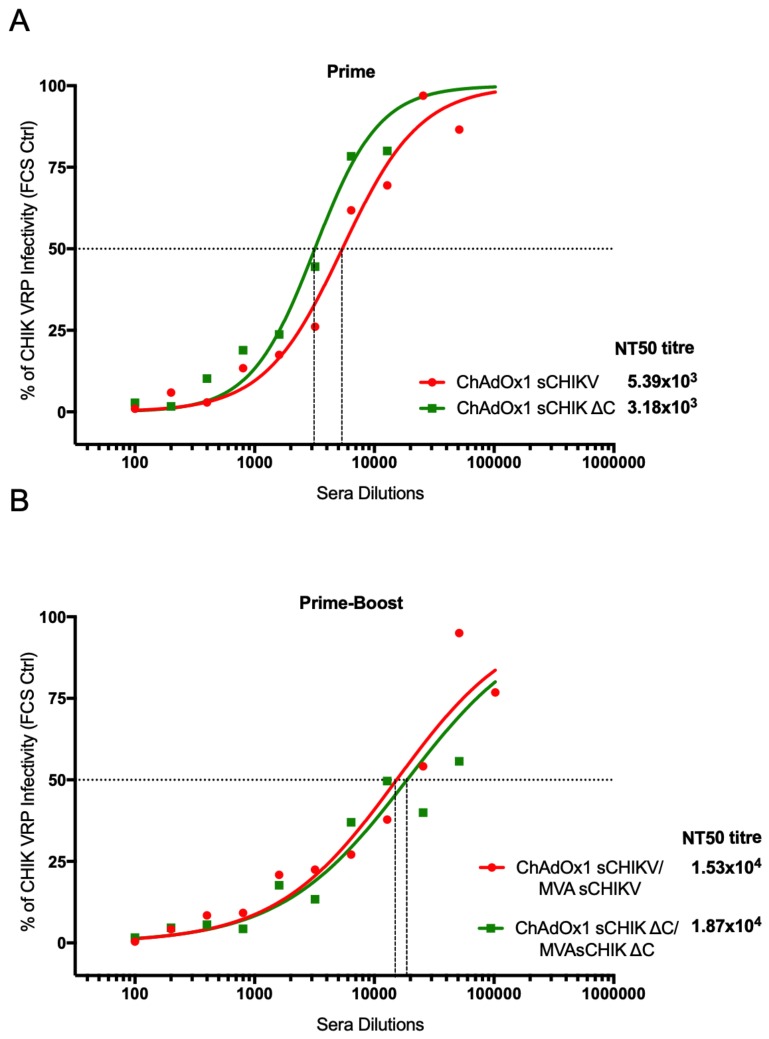
CHIKV neutralisation assay. NT50 titres were assessed upon incubation of BHK-21 cells with CHIKV virus replicon particle-based (VRP) in the presence or absence of serially diluted sera from BALB/c mice, vaccinated with either ChAdOx1 sCHIKV (red line) or ChAdOx1 sCHIKV ∆C (green line) vaccines at 2-week post prime (**A**) and 1-week post-boost (**B**). NT50 titres were calculated using a non-linear fit of the log-dose versus response.

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
