# Peer review of "Assessment of Immunogenicity and Neutralisation Efficacy of Viral-Vectored Vaccines Against Chikungunya Virus"

_viruses, 2019, doi:10.3390/v11040322_

Round 1
Reviewer 1 Report
The manuscript by López-Camacho et al., (Manuscript number: viruses-465351) entitled "Assessment of immunogenicity and neutralisation efficacy of
viral-vectored vaccines against Chikungunya virus" performs viral-vectored vaccines containing all the CHIKV structural proteins or lacking protein C, and shows how the prime vaccination with vaccines based on Chimpanzee adenoviral vectors (ChAdOx1 CHIKV) or the heterologous prime-boost vaccination with vaccines based on ChAdOx1 CHIKV and Modified Vaccinia Ankara (MVA) vector induces CHIKV-specific cellular and humoral responses and neutralization activity in mice sera. The manuscript is in general well written, and the experiments are well designed, however there are some points that need to be completed:
Figure 1 A, B, and C are of poor quality and difficult to read. Please improve quality.
Some references are missing in the text (Lines 234, 309, 314 and elsewhere)
In CHIKV-humoral responses the results of the groups ChAdOx1-ΔC/MVA CHIKV or ChAdOx1-ΔC/MVA CHIKV-ΔC are not shown. It would be desirable to show them and discuss possible differences produced by deletion of Capsid protein.
Similarly as previously mentioned, neutralizing capacity of ChAdOx1-ΔC/MVA CHIKV or ChAdOx1-ΔC/MVA CHIKV-ΔC prime-boost groups is not assessed, making difficult comparisons between groups of ChAdOx1-ΔC prime vaccinated sera with ChAdOx1/MVA CHIKV-ΔC, as prime vaccination was made with different vaccine platforms. On the other hand, results of 6 weeks prime vaccination are not shown, being more suitable data for comparisons with prime-boost vaccination . Please show all these experiments and discuss the new results.
Line 318. Correct Figure 4b instead of Figure 4a.
Please check and homogenize “titer” spelling along the text.
Author Response
Figure 1 A, B, and C are of poor quality and difficult to read. Please improve quality.
Response: Dear Reviewer, thank you for your feedback.
• We have re-formatted the figure 1 (page 6) to include only the tree and the diagrams of
conservation analysis.
• In order to make the figure clearer we have added an extra figure (figure 2, page 7), which
focuses only in the vaccine designs. We have also made figure 2 more comprehensive to
show the genomic organisation of our vaccine constructs.
Some references are missing in the text (Lines 234, 309, 314 and elsewhere)
Response: Thank you for highlighting the missing references
• We now have added the reference in line 258 (previously line 234)
• Apologies; our mistake here; there should be no reference in this sentence in line 325 (
previously 309)
• We now have added the reference in line 340 (previously line 314)
• We have added the reference to line 416
In CHIKV-humoral responses the results of the groups ChAdOx1-ΔC/MVA CHIKV or ChAdOx1-
ΔC/MVA CHIKV-ΔC are not shown. It would be desirable to show them and discuss possible
differences produced by deletion of Capsid protein.
Response: Dear Reviewer, thank you for your comment. Upon suggestions from the rest of the reviewers, we have now a clearer nomenclature for the vaccines presented in this manuscript, as follows:
• ChAdOx1 sCHIKV : adenovirus expressing the structural cassette (s) from CHIKV.
• ChAdOx1 sCHIKV ΔC: adenovirus expressing the structural cassette (s) from CHIKV with
deletion of capsid (ΔC)
• ChAdOx1 NS: Adenovirus expressing non-structural regions (NS). This is a control vaccine.
• MVA sCHIKV: Modified Vaccinia Ankara expressing the structural cassette (s) from CHIKV.
• MVA sCHIKV ΔC: Modified Vaccinia Ankara expressing the structural cassette(s) from
CHIKV with deletion of capsid (ΔC)
• MVA NS: Modified Vaccinia Ankara expressing non-structural regions (NS). This is a control
vaccine.
Regarding your comment, we actually have shown the responses from the prime-boost group
ChAdOx1 sCHIKV ΔC/MVA sCHIKV-ΔC (Figure 5b). However, we did not assess an antigen
heterologous immunisation using sCHIKV and sCHIKV-ΔC antigens, and vice versa. We agree it
would be a very interesting venue to explore which will involve a robust amount of experiments and
animal work and. Although this hypothesis is beyond this study perhaps this would be a project for
one of our students, as we did not consider this exploratory experiment before. Thank you for your
suggestion.
Similarly as previously mentioned, neutralizing capacity of ChAdOx1-ΔC/MVA CHIKV or ChAdOx1-
ΔC/MVA CHIKV-ΔC prime-boost groups is not assessed, making difficult comparisons between
groups of ChAdOx1-ΔC prime vaccinated sera with ChAdOx1/MVA CHIKV-ΔC
Response: Dear Reviewer, thank you for this observation. We would like to confirm that we did show the NT capacity of the group ChAdOx1-CHIKV ΔC/MVA CHIKV-ΔC. (shown in Figure 6, page 13). However, we did not assess the heterologous antigen vaccination ChAdOx1- CHIK ΔC/MVA CHIK which relates to the same justification as our previous comment.
On the other hand, results of 6 weeks prime vaccination are not shown, being more suitable data for comparisons with prime-boost vaccination . Please show all these experiments and discuss the new results.
Response: Dear Reviewer, thank you for your feedback. The objective of our vaccines is to generate functional immune responses when using our designed antigen candidates. We have shown the important observation that our vaccines are capable to generate neutralising antibodies after a short period of time. Vaccines should be aimed to guarantee protection in a shortest window of time between vaccination and exposure to the pathogen, (i.e. a tourist getting a vaccine 2 weeks before traveling to an endemic CHIKV area). Therefore, this is the reason we focused on such a short interval of time. However, we strived to do our best to incorporate reviewer’s comments and your feedback; we now have included an important time-point. In the revised version, we are including the ELISA results for mice vaccinated with a single ChAdOx1 immunisation after 10 months (Figure 5A, bottom and Figure 5C). As it can be appreciated, the titres of antibodies are remarkably maintained over such a long period of time, similar to the 6-week prime. As you can appreciate, the levels of antibodies at 2 weeks after prime are lowest in comparison to those from 6 weeks or 10 months. Therefore, it is expected that at 6 weeks prime or later, we will find evidence of neutralisation; as neutralisation was also measured in the prime-boost. Unfortunately, we find it extremely difficult to repeat the NT experiments at later time-points due to funding and human-resource capabilities.
Line 318. Correct Figure 4b instead of Figure 4a
Response: Thank you for your feedback. We have corrected this (Lines 329 and 330). Figure 4 becomes Figure 5. As we added figure 2 to show a more comprehensive representation of our viral-vectored vaccines. Thank you.
Please check and homogenize “titer” spelling along the text.
Response: Dear reviewer, we have now homogenised the spelling for it throughout the paper.
Thanks for highlighting this issue.
Reviewer 2 Report
This manuscript by López-Camacho and colleagues describes development and characterization of a chimp adenoviral-vectored vaccine for chikungunya virus, as well as a corresponding MVA-based vaccine. The manuscript is clearly presented, comprehensive in terms of characterizing expression and both T and B cell responses, and the work appears to have been expertly conducted. I really have no issues and criticisms of the science and applaud the authors for a nice study. There are a number of minor editorial problems, dominated by aberrant capitalization. Examples include:
Line 70: Live attenuated -> live attenuated, Chimeric -> chimeric
Line 71: Viral Like Particles -> virus like particles
Line 80: Chikungunya -> chikungunya
Line 98: Non structural -> non-structural
Line 158: Anti-mouse -> anti-mouse
The authors are encouraged to scrutinize the entire manuscript for such errors. Some addition minor comments include:
Line 64: million -> million reported cases
Line 79: add comma after ChAdOx1
Line 94 and following: indicate the route of immunization employed
Line 140: 3 days -> Three days
Line 309: missing “ref”
Line 329: impaired -> impair
Line 343: CHIK Virus -> CHIKV
Line 398: to express -> of expressing
Line 446: several vaccine development -> several vaccine development efforts
Author Response
This manuscript by López-Camacho and colleagues describes development and characterization of a chimp adenoviral-vectored vaccine for chikungunya virus, as well as a corresponding MVA-based
vaccine. The manuscript is clearly presented, comprehensive in terms of characterizing expression
and both T and B cell responses, and the work appears to have been expertly conducted. I really
have no issues and criticisms of the science and applaud the authors for a nice study. There are a
number of minor editorial problems, dominated by aberrant capitalization. Examples include:
Response:Dear reviewer, thanks for highlighting this issue. We have now corrected the capitalisation throughout the paper.
We would like to let you know that upon suggestions from the rest of the reviewers, we have now a
clearer nomenclature for the vaccines presented in this manuscript, as follows:
• ChAdOx1 sCHIKV : adenovirus expressing the structural cassette (s) from CHIKV.
• ChAdOx1 sCHIKV ΔC: adenovirus expressing the structural cassette (s) from CHIKV with
deletion of capsid (ΔC)
• ChAdOx1 NS: Adenovirus expressing non-structural regions (NS). This is a control vaccine.
• MVA sCHIKV: Modified Vaccinia Ankara expressing the structural cassette (s) from CHIKV.
• MVA sCHIKV ΔC: Modified Vaccinia Ankara expressing the structural cassette(s) from
CHIKV with deletion of capsid (ΔC)
• MVA NS: Modified Vaccinia Ankara expressing non-structural regions (NS). This is a control
vaccine.
Line 70: Live attenuated -> live attenuated, Chimeric -> chimeric
Response: Thank you, it is now corrected (line 71)
Line 71: Viral Like Particles -> virus like particles
Response: Thank you, it is now corrected
Line 80: Chikungunya -> chikungunya
Response: Thank you, it is now corrected
Line 98: Non structural -> non-structural
Response: Thank you, it is now corrected (line 129)
Line 158: Anti-mouse -> anti-mouse
Response: Thank you, it is now corrected (line 161)
The authors are encouraged to scrutinize the entire manuscript for such errors. Some addition minor comments include:
Line 64: million -> million reported cases
Response: Thank you, it is now corrected
Line 79: add comma after ChAdOx1
Response:Thank you, it is now corrected
Line 94 and following: indicate the route of immunization employed
Response:Thank you, it is now corrected (line 135)
Line 140: 3 days -> Three days
Response:Thank you, it is now corrected (line 144)
Line 309: missing “ref”
Response: Apologies; our mistake here; there should be no reference in this sentence in line 325 ( previously 309)
Line 329: impaired -> impair
Response: Thank you, it is now corrected (line 338)
Line 343: CHIK Virus -> CHIKV
Response: Thank you, it is now corrected (line 363)
Line 398: to express -> of expressing
Response: Thank you, it is now corrected (line 421)
Line 446: several vaccine development -> several vaccine development efforts
Response: Thank you, it is now corrected (line 472)
Reviewer 3 Report
This is a very important work addressing an unmet need in the public health field. The work is very well performed. Vaccine candidates are well characterized before the immunization experiments. However there some issues that need to be corrected in order to improve the message of this
work.
Major findings:
It is difficult to follow the timing of the vaccine regime and the nomenclature the authors proposed.
Under Materials and Methods, immunization of animals there is not mention about the timing of the boost. It is described first time under CHIKV-humoral responses after vaccination.
In addition, under that same section and throughout the text authors use different nomenclature defining the vector.
ChAdOx1 NS
MVA NS
ChAdOx1 CHIKV
ChAdOx1 CHIKV-ΔC
MVA CHIKV]
MVA CHIKV-ΔC
MVA NS
ChAdOx1/MVA NS
ChAdOx1/MVA CHIKV-ΔC
However, the nomenclature does not correspond with the names assigned to the groups in the figures. This discrepancy made very difficult to follow up the results.
Also, in figure 4 C and D have the same legend as A and B. However as per the text all animals were challenged with some combination but that challenged is not described in the legend.
Same combination of different nomenclatures are used in the discussion which made difficult to follow it.
Minors
Figure 1D simplify the vector description and shows ChAdOx1 and MVA together.
May be authors would like to consider taking some time to push the return key and to provide a name for each construct.
Additionally, the diagram for the other vectors mentioned and used in the work should be presented (ex. ChAdOx1 NS and MVA NS).
Antibodies titers are presented 2 weeks after 2 and 6 weeks post prime and two weeks postboost (Figure 4). While neutralisation titers are presented at 2-week post-prime and 1-week post-boost (Figure 5).
It would be interesting to know why the authors choose to provide the antibodies and neutralization results from different time points and not from all or the same time point.
Authors missed adding a reference in line 314.
Author Response
This is a very important work addressing an unmet need in the public health field.
Response: Dear reviewer, we appreciate your comment, you are right. The need of a vaccine is urgent and unfortunately there is no licenced vaccine for CHIKV yet. We would like to let you know that we have are now running a Phase I clinical trial for our ChAdOx1 sCHIKV vaccine in healthy volunteers here at Oxford. Preliminary data show that volunteers present seroconversion and neutralisation titters. We hope to report the results as soon as the trial is finished. Thank you for your feedback.
The work is very well performed. Vaccine candidates are well characterized before the immunization experiments. However there some issues that need to be corrected in order to improve the message of this work.
Major findings:
It is difficult to follow the timing of the vaccine regime and the nomenclature the authors proposed.
Response: Dear reviewer, now we have made both the description and nomenclature of our vaccines in a clearer way, thanks for your feedback and for highlighting the issue.
Under Materials and Methods, immunization of animals there is not mention about the timing of the boost. It is described first time under CHIKV-humoral responses after vaccination. In addition, under that same section and throughout the text authors use different nomenclature defining the vector.ChAdOx1 NS MVA NS ChAdOx1 CHIKV ChAdOx1 CHIKV-ΔC MVA CHIKV] MVA CHIKV-ΔC MVA NS ChAdOx1/MVA NS ChAdOx1/MVA CHIKV-ΔC
Response: • Dear reviewer, now we have included a description in the figure legend 5.
• Nomenclature of our vaccines is being corrected/ homogenised and shown in a clearer way, thanks for your feedback and for highlighting the issue
However, the nomenclature does not correspond with the names assigned to the groups in the figures. This discrepancy made very difficult to follow up the results.
Response: Dear reviewer, we would like to let you know that upon suggestions from you and the rest of the reviewers, we have now a clearer nomenclature for the vaccines presented in this manuscript.
• ChAdOx1 sCHIKV : adenovirus expressing the structural cassette (s) from CHIKV.
• ChAdOx1 sCHIKV ΔC: adenovirus expressing the structural cassette (s) from CHIKV with deletion of capsid (ΔC)
• ChAdOx1 NS: Adenovirus expressing non-structural regions (NS). This is a control vaccine.
• MVA sCHIKV: Modified Vaccinia Ankara expressing the structural cassette (s) from CHIKV.
• MVA sCHIKV ΔC: Modified Vaccinia Ankara expressing the structural cassette(s) from CHIKV with deletion of capsid (ΔC)
• MVA NS: Modified Vaccinia Ankara expressing non-structural regions (NS). This is a control vaccine. Apologies for the confusion.
Thank you.
Also, in figure 4 C and D have the same legend as A and B. However as per the text all animals were challenged with some combination but that challenged is not described in the legend. Same combination of different nomenclatures are used in the discussion which made difficult to follow it.
Response: Dear reviewer, thanks for your comment. Apologies for the misunderstanding. We would like to confirm that we are not presenting a challenge experiment in this manuscript. However, we are presenting evidence of functional antibodies in an in vitro neutralisation experiment. However, we
have strived to make this clear in the manuscript, stating that figure 5 (previously 4) corresponds to ELISAS only. Whereas figure 6 (previously 5) correspond to in vitro neutralisation.
Minors
Figure 1D simplify the vector description and shows ChAdOx1 and MVA together.
Response: Dear reviewer, we have now improved the figure and present it as an additional figure (figure 2), with the hope to make a clearer representation of our vaccine candidates. Thanks for this important comment.
May be authors would like to consider taking some time to push the return key and to provide aname for each construct. Additionally, the diagram for the other vectors mentioned and used in the work should be presented (ex. ChAdOx1 NS and MVA NS).
Response: Dear reviewer, thanks; we have now improved the nomenclature. We have now modified the figure and figure legends to reflect the changes in the manuscript.Thank you very much for your time at highlighting this problem.
Antibodies titers are presented 2 weeks after 2 and 6 weeks post prime and two weeks postboost (Figure 4). While neutralisation titers are presented at 2-week post-prime and 1-weekpost-boost (Figure 5). It would be interesting to know why the authors choose to provide the antibodies and neutralization results from different time points and not from all or the same time point.
Response: Dear reviewer, thanks for your comment. You are correct about this observation. We have seen that upon a booster immunisation, similar antibody titres are raised at one- or two-weeks post-boost. Therefore, timing here was not a crucial variable and it does not affect the fact that our vaccines are capable to mount neutralising antibodies against CHIKV in vitro.
Based on our Home Office government licence for animal experimentation, we are restrained to obtain large amount of sampling when tail-bleeding animals. Since we had to ship sera to our collaborators in Germany or elsewhere, the availability of sera was a key factor and hence the variability of samples. We would like to stress that this eventuality does not change the fact in any way that our vaccines are immunogenic in mice and that we are presenting evidence of neutralisation in vitro. We believe, we are accomplishing the aim of our manuscript to demonstrate that our novel design is actually generating functional efficacy in vitro. Thank you for your time at reading our work, it is highly appreciated.
Authors missed adding a reference in line 314
Response: Thank you, it is now corrected
Round 2
Reviewer 1 Report
All my previous concerns about the manuscript have been clarified or corrected. Just to point out that in figure 2 B in ChAdOx1 sCHIKV, a ΔC is missing.
the authors properly addressed the critics.
Reviewer 3 Report
At this point, at least from my perspective, the manuscript can be accepted for publication.
Minor point: figures orders is messy. But I guess the editorial board will help on this.